# Effect of SARS-CoV-2 Mutations on the Efficacy of Antibody Therapy and Response to Vaccines

**DOI:** 10.3390/vaccines9080914

**Published:** 2021-08-17

**Authors:** Ahmed Yaqinuddin, Areez Shafqat, Junaid Kashir, Khaled Alkattan

**Affiliations:** 1College of Medicine, Alfaisal University, Riyadh 11533, Saudi Arabia; ashafqat@alfaisal.edu (A.S.); jkashir@alfaisal.edu (J.K.); KKattan@alfaisal.edu (K.A.); 2Department of Comparative Medicine, King Faisal Specialist Hospital and Research Center, Riyadh 11533, Saudi Arabia

**Keywords:** mutations, SARS-CoV-2, vaccines, monoclonal antibodies, resistance

## Abstract

SARS-CoV-2 causes severe acute respiratory syndrome, which has led to significant morbidity and mortality around the world. Since its emergence, extensive prophylactic and therapeutic countermeasures have been employed to successfully prevent the spread of COVID-19. Extensive work has been undertaken on using monoclonal antibody therapies, mass vaccination programs, and antiviral drugs to prevent and treat COVID-19. However, since antiviral drugs could take years to become widely available, immunotherapy and vaccines currently appear to be the most feasible option. In December 2020, the first vaccine against SARS-CoV-2 was approved by the World Health Organization (WHO) and, subsequently, many other vaccines were approved for use by different international regulators in different countries. Most monoclonal antibodies (mAbs) and vaccines target the SARS-CoV-2 surface spike (S) protein. Recently, mutant (or variant) SARS-CoV-2 strains with increased infectivity and virulence that evade protective host antibodies present either due to infection, antibody therapy, or vaccine administration have emerged. In this manuscript, we discuss the different monoclonal antibody and vaccine therapies available against COVID-19 and how the efficacy of these therapies is affected by the emergence of variants of SARS-CoV-2. We also discuss strategies that might help society cope with variants that could neutralize the effects of immunotherapy and escape the protective immunity conferred by vaccines.

## 1. Introduction

Coronavirus disease 2019 (COVID-19), caused by severe acute respiratory syndrome coronavirus 2 (SARS-CoV-2), was declared a pandemic on 11 March 2020. Since then, several prophylactic and therapeutic countermeasures have been intensely scrutinized. Antibody therapy, mass vaccination programs, and antiviral drugs are regarded as the most effective measures. However, even though numerous antiviral drugs are currently in advanced testing, the development of new drugs is complex and can be a prolonged process, with repurposed drugs currently representing an alternative [1]. However, while repurposing current drugs against SARS-CoV-2 is promising, viral plasticity limits their utility [2]. Therefore, while recent efforts are indeed encouraging, it may take some time for suitable and widely effective antiviral drugs to become globally available. Until then, vaccination appears to be the most feasible option.

Most monoclonal antibodies (mAbs) and vaccines target the SARS-CoV-2 surface spike (S) protein [3,4]. However, mutant SARS-CoV-2 strains with increased infectivity and virulence may evade protective host antibodies present due either to infection, antibody therapy, or vaccine administration. Such variants could neutralize the effects of immunotherapy and escape the protective immunity conferred by vaccines.

Here, we discuss contemporary SARS-CoV-2 mutant strains that hamper the efficacy of current vaccines and antibody therapies. Indeed, continuous monitoring for new SARS-CoV-2 sequences based on which mAbs and modifications to vaccines should be developed is an ongoing focus for most public health authorities. Future mRNA and viral-vector COVID vaccines must also be effective against prevalent SARS-CoV-2 variants. Hypothetically speaking, this could suggest that live attenuated vaccines or nanoparticle-based vaccines (nanovaccines) are more effective and convenient due to their greater stimulation of immune response. Approaches to enhance host defense in the setting of SARS-CoV-2 variant antibody escape include administering antibody cocktails as prophylaxis or therapy.

## 2. Neutralizing Antibodies

Monoclonal antibodies (mAbs) antagonizing the interleukin-6 (IL-6) receptor have been utilized as anti-inflammatory agents to combat the cytokine storm characteristic of severe COVID-19. In this context, the mAbs studied include tocilizumab, sarilumab, and siltuximab [5]. However, the results from such investigations were inconclusive. More efficacious results were observed with the administration of convalescent sera of recovering COVID-19 patients, which improved the overall clinical status of critically ill COVID patients upon administration due to their high titers of neutralizing antibodies (nAbs) [6,7].

The entry of SARS-CoV-2 into human cells is mediated by the interaction of surface spike (S) proteins and angiotensin-converting enzymes (ACE2), after which the S protein is primed by transmembrane serine protease 2 (TMPRSS2) [8]. The S protein has two subunits (S1 and S2), which facilitate viral entry. S1 contains the receptor-binding domain (RBD) and interacts with ACE2, while S2 allows subsequent membrane fusion and cell entry [9,10]. Several neutralizing antibodies targeting the RBD of S protein are being analyzed and evaluated clinically [11,12]. The N terminal-domain (NTD) of S protein also contains immunoprotective epitopes, the most immunodominant of which elicits the production of antibody 4A8, a mAb with high neutralizing potency against the NTD of SARS-CoV-2 [13,14].

Bamlanivimab and Etesevimab, codenamed LY-CoV555 and LY-CoV016, respectively, were granted emergency use authorization (EUA) by the US Food and Drug Administration (FDA) on 10 November 2020, specifically for nonhospitalized COVID-19 patients with mild to moderate infection but with a high risk of progressing to severe disease or hospitalization [15]. Bamlanivimab, approved as a mAb monotherapy by the FBA, is a human neutralizing IgG_1_ that targets the RBD of the SARS-CoV-2 S protein to block attachment to ACE2. Etesevimab is a mAb that targets an epitope of the S protein distinct from that of Bamlanivimab and was developed for synergistic effect with Bamlanivimab [16]. A BLAZE-1 study evaluating the efficacy of both bamlanivimab monotherapy and combination therapy (bamlanivimab/etesevimab) compared to a placebo revealed significant reductions in SARS-CoV-2 load with the cocktail [16]. Another phase 2 clinical trial assessing the impact of 700 mg, 2800 mg, and 7000 mg doses of bamlanivimab monotherapy on quantitative viral load revealed significant decreases in viral load in patients receiving the 2800 mg dose, whereas the other two doses did not accelerate the decrease in viral load [17].

Similarly, REGN-COV2 is an antibody cocktail comprised of two non-competing neutralizing mAbs named imdevimab (REGN10987) and casirivimab (REGN10933). It was approved for EUA by the FDA for use in the same setting as Bamlanivimab and Etesevimab. Imdevimab and casirivimab both target the S protein RBD, preventing virus–host cell interaction, thus leading to neutralization. The binding site of imdevimab overlaps significantly with that of ACE2, whereas the engagement of casirivimab to its receptor does not obstruct ACE2 binding. The REGN-COV2 antibody cocktail was shown to reduce viral load in the respiratory tract and, consequently, COVID-19 severity and complications in both rhesus macaques and golden hamsters [18]. Provisional results and analysis of data involving 275 nonhospitalized COVID-19 patients in an ongoing double-blind, phase 1–3 trial evaluating the efficacy and safety of REGN-COV2 against placebo showed significant reductions in viral loads in REGN-COV2 receivers [19,20]. Furthermore, no mutant strains arose secondary to combination therapy with imdevimab and casirivimab. Cocktail therapy is therefore preferred over monotherapy [21].

Antibodies targeting the viral nucleocapsid (N) protein—which may mediate viral genome expression and assembly—have been isolated from convalescent sera of COVID-19 patients in the early recovery phase; in contrast, anti-S protein antibodies dominate late-phase sera [14,22,23]. A recent study highlighted a particular anti-N-protein nAb, nCoV396, which inhibited N-protein–MASP-2 interaction [14], thus blocking N-protein-mediated complement hyperactivation and combatting the pro-inflammatory state characterizing COVID-19 [14]. Further work is required to elucidate the full prophylactic and therapeutic potential of these Abs.

## 3. Types of COVID-19 Vaccines

Vaccines aim to trigger the adaptive arm of host immunity—comprising both cell-mediated and humoral responses—leading to the generation of neutralizing, or virus-blocking, antibodies. Vaccines contain an antigen of the virus whose introduction triggers antibody production, which confers protective immunity. In the context of SARS-CoV-2, the antigen administered is the surface S protein. Different types of COVID vaccines introduce the S protein in different ways to initiate host immune defense without causing disease. On this basis, there are currently four types of COVID vaccines in use: viral vector vaccines, genetic vaccines using nucleic acids, virus vaccines, and protein vaccines.

Viral vector vaccines use an unrelated virus that infects cells and subsequently expresses the SARS-CoV-2 S protein gene to activate the immune response. The Oxford AstraZeneca (ChAdOx1) and Johnson & Johnson’s (J&J) Janssen (Ad26.COV2.S) vaccines are examples of viral vector vaccines that are currently in use. COVID-19 nucleic acid vaccines contain DNA/RNA encoding the surface S protein. After administration, through transcription and translation, host cells present S protein in the context of major histocompatibility complexes (MGC) to helper and cytotoxic T cells to provide immunity. Examples of this vaccine group are the Pfizer BioNTech (BNT162b2) and Moderna (mRNA-1273) vaccines, which contain RNA encoding the S protein. The third set of vaccines involves injecting a killed/inactivated or weakened form of SARS-CoV-2 to elicit an immune response without causing COVID-19. The Sinopharm and Sinovac vaccines are examples of inactivated vaccines. Currently, the Pfizer-BioNTech, Moderna, and J&J vaccines have gained approval by the FDA for use in the United States of America.

Protein vaccines inject viral proteins, such as the S protein of SARS-CoV-2, into the body to elicit an immune challenge. One such vaccine under development is the Novavax vaccine, codenamed NVX-CoV2373. Comprising a full-length S protein on a nanoparticle platform that is administered intramuscularly with a matrix-M adjuvant, it can trigger a protective Th1-dominant B and T cell response in mice and baboons [24]. A placebo-controlled phase 1–2 trial involving 131 healthy adults compared the reactogenicity and safety of anti-S protein IgG titers generated from the administration of NVX-CoV2373 to convalescent serum from symptomatic COVID-19 patients. The adjuvanted two-dose regimen triggers a Th1 dominant immune response that demonstrated a four times higher efficacy compared to convalescent sera, reaching levels comparable with those observed in hospitalized COVID-patients [25]. Thus, nanovaccines, such as virus-like particles (VLPs), may also represent a viable and rapid approach, serving as ideal scaffolds for antigen display due to their emulation of naturally occurring viruses in size and geometry [26,27,28,29]. This results in an enhanced clustering of B cell receptors and resultant immunogenicity [30]. Indeed, the S protein displayed on nanoscale scaffolds seems more immunogenic in mice compared to S protein administered alone [31]. In Syrian hamsters and mice with SARS-CoV-2, this approach generated high neutralizing antibody titers following a single dose, with no infectious virus detected in the lungs [29,32].

## 4. Efficacy of Vaccines, Convalescent Sera, and Neutralizing Antibodies on SARS-CoV-2 Variants

SARS-CoV-2 acquires mutations that lead to genetic drift due to changes in infectivity and virulence, antigenicity, and the ability to evade pre-existing host antibodies acquired either through infection, antibody therapy, or vaccination. Initially, SARS-CoV-2 underwent genetic evolution, giving rise to the D614G variant that subsequently became the dominant variant during the pandemic owing to its increased transmissibility [25,33]. However, subsequent studies have revealed that the D614G mutation might confer increased susceptibility to neutralization by antibodies and, therefore, does not pose a threat to vaccine development [33].

Since then, many variants have arisen, some of which have been classified by the CDC and WHO as variants of concern (VOCs) and variants of interest (VOIs) (Figure 1). This categorization was based on variants possessing increased transmissibility and virulence, the ability to evade detection, and the ability to curb prophylactic and therapeutic countermeasures such as vaccines, neutralizing mAbs, and the administration of convalescent sera. The currently designated VOCs include four variants, including strains first detected in the UK (B.1.1.7 lineage or alpha), South Africa (B.1.351 lineage or beta), Brazil (P.1 lineage or gamma), and India (B.1.617.2 lineage or delta) [34,35]. The alpha, beta, and gamma variants exhibit a higher transmissibility, which is attributed to the N501Y mutation common to all three strains, as it improves the affinity of S protein to host cell receptors [36,37]. However, the N501Y mutation does not affect neutralization by protective host antibodies and, therefore, does not interfere with the potency of vaccines, convalescent sera, or mAbs. The delta variant (B.1.617 lineage—with 3 subsets) was first described in India in October 2020 and is believed to account for the recent increase in COVID-19 cases. It exhibits characteristic mutations in the S protein, including D11D, G142D, L45R, and E484q. L452R and E484Q are located in the RBD and, along with the P681R mutation in the furin cleavage site at the junction between S1 and S2, are thought to confer increased transmissibility to the delta variant [38,39].

The analysis of these variants revealed that the alpha variant discovered in the UK was resistant to neutralization by most mAbs targeting the NTD of the S protein and relatively resistant to a few mAbs against the RBD. However, responses to convalescent sera or vaccine sera were unchanged compared to ancestral strains [40,41]. By contrast, the beta variant, in addition to displaying E484K and K417N spike mutations, is resistant to neutralization by mAb therapy, convalescent sera, and vaccinee sera from individuals vaccinated with the Pfizer-BioNTech and Moderna mRNA vaccines. The alpha and gamma variants exhibit normal neutralization—relative to the parent strain—with these mRNA vaccines [40,42,43,44]. Therefore, an increase in the prevalence of the beta variant poses a threat to the efficacy of the current mRNA vaccines. In addition, the results from a clinical trial revealed that the Oxford AstraZeneca vaccine fails to confer protection from the beta variant, with an overall efficacy of 22% [45]. Furthermore, the clinical efficacy of the vaccine against the beta variant was reduced relative to ancestral strains [46]. Encouragingly, however, the REGN-COV2 and bamlanivimab/etesevimab antibody cocktails retained their activity in vitro against the alpha and beta VOCs.

A phase 3 trial conducted in South Africa evaluating the efficacy of the J&J vaccine involving 6576 participants revealed efficacies of 52% and 64% at days 14 and 28, respectively [47]. The efficacy of the two-dose adjuvanted regimen of NVX-CoV2373 Novavax is currently being assessed in an ongoing randomized, observer-blinded, placebo-controlled 2a-b trial involving 6324 participants in South Africa. Preliminary safety results were acceptable, with headache (20–25%), muscle pain (17–20%), and fatigue (12–16%) being the most common [48]. Early results estimated a vaccine efficacy of 52.2%. Additionally, this trial showed that infection with the prototypical, wildtype (WT) SARS-CoV-2 Wuhan strain does not protect against reinfection by the beta variant [48]. Therefore, the J&J and Novavax vaccines afford modest protection against beta and gamma VOCs.

The resistance of the B.1.617.1 lineage (a subset of the delta variant) to convalescent sera and nAbs induced by the Pfizer and Moderna vaccines was assessed. Neutralization titers were found to be lower in all relative to the WT; B.1.617.1 was 6.8 times less susceptible to neutralization by convalescent and vaccinee sera [49]. Another study evaluating the degree of inhibition of S protein–ACE 2 interaction and subsequent cell entry by convalescent sera from intensive care unit patients revealed a twofold reduction in inhibition relative to the WT; by comparison, the delta variant showed a sixfold decrease [50]. Additionally, delta showed a threefold reduction in inhibition by sera obtained from individuals who received the Pfizer-BioNTech vaccine compared to WT. In contrast, beta exhibited an elevenfold reduction [50]. Monotherapy with either mAb failed to inhibit virus-host cell interactions [43,50]. However, bamlanivimab/etesevimab cocktail therapy has recently been reported to resolve symptoms in a severely ill patient infected with the delta variant [51]. Furthermore, studies testing the efficacy of the Pfizer-BioNTech and Oxford AstraZeneca vaccines against the delta variant relative to alpha revealed only small differences with two-dose regimens; indeed, one dose barely induced nAbs [52,53]. This suggests that greater public emphasis be placed on two-dose vaccinations in susceptible populations for protection against the delta (B.1.617.2) variant [53].

It is important to note that these studies have not yet been formally peer-reviewed and, consequently, are not definitive. Therefore, the results of these preliminary studies should be interpreted with caution because all vaccines did show efficacy, albeit to varying success, and an objective cut-off value for a protective antibody response is currently not known. Furthermore, COVID-19 vaccines elicit antibody production against various, unrelated S protein epitopes, allowing for the possibility of some nAbs retaining neutralizing capacity against the VOCs. Lastly, other components of adaptive host defense, such as T cell-mediated immunity, may constitute important mechanisms unaffected by the VOCs. Rather encouragingly, concerns underlying variants that could be partially resistant to vaccine-induced antibody responses may exhibit other immune responses that protect against viruses. Of particular interest is T cell-mediated immunity, which targets and eliminates virus-infected cells. Indeed, data are increasingly suggesting that such responses could provide enhanced immunity against COVID-19 despite decreased effectiveness of humoral responses [54,55,56]. While T cells do not prevent infections, they exert a role in clearing them, potentially meaning the difference between a mild or severe infection, and also potentially reducing community transmission [54,56]. Furthermore, T cells could also be more resistant to the genetic drift of emerging variants compared to antibodies [54]. Indeed, T cell responses to coronavirus vaccination or previous infection did not target mutation-susceptible regions of SARS-CoV-2, conferring a type of ‘resistance’ against such mutations [54,56]. While most antibody-based vaccines target the mutation-prone S region, T cells target very stable viral proteins inside infected cells, potentially indicating powerful avenues of vaccination research incorporating targets from multiple proteins into one vaccine.

## 5. Perspectives and Future Directions

In light of these findings and the likelihood of future antigenic drift in the S protein, we propose that the development of future mRNA and viral vector vaccines should include various, prevalent S protein mutant sequences. Alternatively, in cases of reduced vaccine efficacy, antibody cocktail administration as prophylaxis and therapy should be the subject of future research. Lastly, considering future potentially resistant strains, live attenuated vaccines could theoretically constitute better options than mRNA and viral vector vaccines from a convenience standpoint. However, the live virus requires specific storage conditions, limiting its widespread use in low-resource countries. Additionally, live-attenuated vaccines suffer from well-established safety concerns; indeed, reversion to virulence precludes the use of such vaccines in the elderly and immunocompromised. Although in such cases the current Novavax and Sinopharm/Sinovac vaccines could be attractive options, such subunit and inactivated vaccines may not be optimal due to low immunogenicity. Indeed, clinical trials evaluating the efficacy of the inactivated Sinopharm and Sinovac vaccines have revealed a lower efficacy than the mRNA vaccines against the Wuhan reference strain as well as the VOCs [53,57]. However, due to favorable storage conditions (2–8 degrees Celsius), these vaccines are in widespread use in low-to-middle-income countries [58].

However, nanoparticle vaccines (also termed nanovaccines) could be more feasible from an efficacy and safety standpoint. Indeed, numerous antigens or epitopes can be incorporated into nanoparticles via either entrapment (nanoparticle engulfs and thereby protects the antigen) or conjugation (covalent linkage). Studies have demonstrated such formulations as exhibiting prolonged antigen presentation to immune cells and, consequently, robust immunogenicity [59]. Furthermore, certain nanoparticles, such as carbon nanotubes, carbon black nanoparticles, silicon dioxide, titanium dioxide, etc., independently act as adjuvants by inducing reactive oxygen species production in APCs, leading to NLPR3 inflammasome activation [60] and the consequent augmentation of the immune response [61,62,63,64]. Indeed, a preprint study demonstrated a polymerase-based nanovaccine delivering the SARS-CoV-2 S protein RBD as eliciting robust immunological responses [65]. In short, nanoparticle-based vaccines show no compromise in immunogenicity, favoring their use in vaccines.

The main advantage of nanoparticles is that due to their size and ability to incorporate viral antigens, these vaccines can mimic respiratory viruses and their pathophysiology. Further, recent advances in this field have enabled the tailoring of nanoparticles with specific shapes, solubility, surface chemistry, and hydrophilicity [66,67]. SARS-CoV-2 and other respiratory viruses are transmitted via droplets and/or aerosols, subsequently multiplying and invading the respiratory mucosa, all while simultaneously propagating along the conducting airways (ciliated respiratory epithelial cells) [68,69]. The current COVID vaccines are administered IM which, while eliciting a systemic IgG immune response, does not induce protective IgA mucosal antigen-specific antibodies [70]. These secretory IgA dimers are 15 times more potent than serum monomeric IgA, highlighting their potential importance as neutralizers of SARS-CoV-2 [71]. Theoretically, despite reducing infection severity, IM vaccine administration does not induce mucosal dimeric IgA and, therefore, does not prevent infection.

On the other hand, nanoparticle-based vaccines do not suffer from such limitations and can mimic virus behavior. Indeed, intranasal routes of administration confer better protection on both systemic and mucosal levels [72,73,74,75,76]. Consistently, studies evaluating the efficacy of nanovaccines against respiratory viruses—such as influenza, respiratory syncytial virus, and Bovine parainfluenza 3—have yielded encouraging results (refer to [67] for a detailed review). Experiments on macaques suggest that although IM administration of the Oxford–AstraZeneca vaccine confers protection against infection, it does not reduce nasal shedding when compared to its effect on control macaques [77]. In comparison, an intranasal construct of the AstraZeneca vaccine reduced nasal shedding in macaques as well as conferring greater protection against transmitted SARS-CoV-2 [77]. Therefore, it is possible that intranasal administration constitutes the solution to reducing SARS-CoV-2 transmission. Indeed, recent and ongoing experiments on macaques and mice have revealed that adenoviral vector vaccines—if administered intranasally—significantly reduce viral transmission, conferring a type of sterilizing immunity [70,78].

Cross-reactive nAbs—which neutralize multiple betacoronaviruses and other strains of the same virus—have been isolated from humans infected with SARS-CoV. One such antibody is the cross-reactive nAb DH1047 [79]. This could represent an avenue of focus for SARS-CoV-2 vaccines. By inducing cross-reactive nAbs, vaccines could combat other betacoronaviruses, currently emerging SARS-CoV-2 VOCs, and future zoonotically-transmitted coronaviruses. To explore this possibility, a recent study designed a VLP coated with RBD of the SARS-CoV-2 spike protein platformed on ferritin nanoparticles, termed RBD-scNP [79]. The protective immunity conferred by RBD-scNP after IM administration to macaques was assessed against lipid-encapsulated nucleosome-modified mRNA encoding transmembrane protein S-2P, termed S-2P mRNA-LNP; this model was designed to simulate the current FDA-approved mRNA vaccines. Intriguingly, serum nAb titers isolated from the VLPs were significantly higher than in the mRNA model and even higher than in natural SARS-CoV-2 human infection [79]. RBD-scNP vaccinee sera were also more resistant to decreased neutralizing potency against the alpha and beta VOCs, while S-2P mRNA immune sera displayed much greater reductions in efficacy. Cross-reactive DH1047-like Abs were induced by both vaccines, with higher titers being elicited by the RBD-scNP nanovaccine; these magnitudes were even higher than those seen in Pfizer-BioNTech and natural SARS-CoV-2 infection sera. Lastly, the RBD-scNP vaccine also elicited RBD-specific mucosal antibodies [79]. Similarly, another study evaluating the efficacy of adjuvanted RBD-conjugated nanovaccines revealed significant nAb and CD4 Th cell responses, with mild reductions in efficacy against the beta variant [80]. Accordingly, these promising results have resulted in two such adjuvanted nanovaccines currently undergoing clinical trials [81,82].

Admittedly, however, several challenges do exist in the production of such vaccines, mainly related to production difficulties [83]. Furthermore, the interactions between nanoparticles as adjuvants and immune cells remain poorly understood and could manifest as potentially untoward adverse effects [67,84]. Nevertheless, nanoparticle-based vaccines merit further investigation and scrutiny as validated strategies. Despite the production difficulties and the fact that current results of mRNA vaccines against VOCs look reassuring, the effectiveness of nanoparticle-based vaccines in combating the aforementioned respiratory viruses could translate over to SARS-CoV-2; intranasally administered nanoparticle-based vaccines could represent a feasible preventative modality to combat the current and any future coronavirus pandemics. Indeed, as discussed above, experimental models of intranasal adenoviral vector vaccines on mice and macaques have yielded highly encouraging results, demonstrating that intranasal administration prevents SARS-CoV-2 infection in the respiratory tract, thus conferring sterilizing immunity. However, we propose that from an immunogenicity perspective, nanovaccines might be superior to such models. Phase I clinical trials of an intranasal adenovirus type 5 vector vaccine (AdCOVID) were recently discontinued based on weak immunogenicity, despite achieving promising preclinical results [85]. Studies comparing the magnitude of immune sera neutralization titers induced by vector and nanovaccines would perhaps provide an answer to this question. Furthermore, the current adenoviral vector COVID-19 vaccines (AstraZeneca and Janssen) have been reported to cause vaccine-induced thrombotic thrombocytopenia in susceptible patients. Multiple studies have postulated this to be mediated by adenovirus-induced platelet activation.

## Figures and Tables

**Figure 1 vaccines-09-00914-f001:**
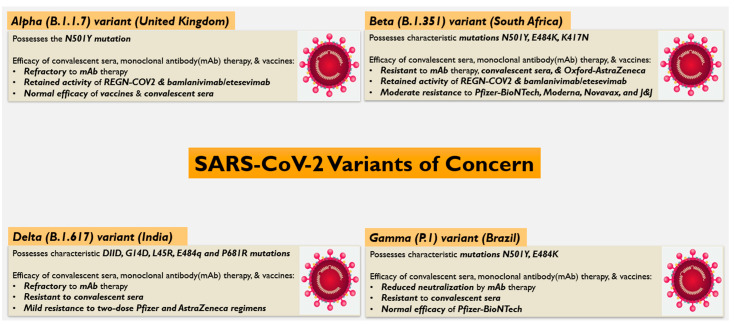
Variants of Concern (VOC): This figure shows the characteristics features of the four major VOCs of SARS-CoV-2: Alpha (United Kingdom), Beta (South Africa), Gamma (Brazil), and Delta (India).

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
