# Peer review of "Effect of SARS-CoV-2 Mutations on the Efficacy of Antibody Therapy and Response to Vaccines"

_vaccines, 2021, doi:10.3390/vaccines9080914_

Round 1

Reviewer 1 Report

The review discusses the monoclonal therapeutical agents available or tested at present and the vaccines available and on the way to approval. It also analyzes the efficascy of vaccination against the UK, Indian, Brasilian and South African variants. The review is comprehensive, interesting and well-written.

I believe an important aspect should be at least briefly discussed, that is T cell immunity.

English: several typos are highlighted in Yellow.

Line 42. “account…did you mean come to terms?

Lines 43-44. I would stress that this sentence is a hypothesis. To my knowledge, so far no attenuated vaccine has proven more effective than the ones available, but in principle this might be true. In addition, why should nanoparticles vaccines be more effective than recombinant or vector based?

Line 49 the definition of antibodies as anti-inflammatory agents is not correct, unless the ones that are not against SARS CoV2 are meant. Please specify.

Line 65 pls explain what Ab 4A8 is.

Line 86. …interaction leading should be interaction, thus leading.

Line 93. Sentence missing the verb.

Line 100. What are isolates of antibodies?

Line 107. Vaccines also stimulate T cells. They are not necessarily made up of a single epitope, I believe the Authors meant antigen not epitope.

Line 132-135. Sentence requires revision.

Fig.1 The variants of interest shown in Fig 1 are classidfied of concern by CDC. Pls insert the delta variant. In addition, please add the Greek letters that are used to identify all variants according to CDC.

Lines 173- 176 verb missing.

Line 206 inhibition by sera…

Lines 241-244. Pls rephrase. Again, why do the Authors believe that these formats of vaccination (especially nanoparticles) should work better? Pls explain.

Author Response

Reviewer 1

1) The review discusses the monoclonal therapeutical agents available or tested at present and the vaccines available and on the way to approval. It also analyzes the efficacy of vaccination against the UK, Indian, Brazilian, and South African variants. The review is comprehensive, interesting, and well-written.

We thank the reviewer graciously for their kind comments and positive view of our article. We hope this will be of benefit to the community at large.

2) I believe an important aspect should be at least briefly discussed, that is T cell immunity.

We thank the reviewer for this highly pertinent suggestion. To address this concern, the following has now been included in the manuscript:

Rather encouragingly, concerns underlying variants that could be partially resistant to vaccine-induced antibody responses may exhibit other immune responses that protect against viruses. Of particular interest is T-cell-mediated immunity, which targets and eliminates virus-infected cells. Indeed, data is increasingly suggesting that such responses could provide enhanced immunity against COVID-19 despite decreased effectiveness of antibody-based responses (Ledford, 2021; Skelly et al., 2021; Tarke et al., 2021). While T cells do not prevent infection, such cells exert a role in clearing an infection, potentially meaning the  difference between a mild or severe infection, potentially also reduce community transmission (Ledford, 2021; Tarke et al., 2021). Furthermore, T cells could also be more resistant to decreased efficacy towards emerging variants compared to antibodies (Ledfor, 2021). Indeed, T cell responses to coronavirus vaccination or previous infection did not target mutated regions found in variants, conferring a type of ‘resistance’ against such mutations (Ledford, 2021; Tarke et al., 2021). While most antibody-based vaccines target the mutation-prone S region of the virus, T cells target very stable viral proteins inside infected cells, potentially indicating powerful avenues of vaccination research incorporating targets from multiple proteins into one vaccine (Ledford, 2021).

3) English: several typos are highlighted in Yellow.

We thank the reviewer for such judicious review of our article. To this degree, we have addressed all identified typos in the manuscript that were highlighted.

4) Line 42. “account…did you mean come to terms?

The reviewer is correct in their thinking. To avoid confusion, we have changed the sentence in question to the following (line 49 in the revised manuscript):

Future mRNA and viral-vector COVID vaccines also must also be effective against prevalent SARS-CoV-2 variants.

5) Lines 43-44. I would stress that this sentence is a hypothesis. To my knowledge, so far no attenuated vaccine has proven more effective than the ones available, but in principle this might be true.

To address this concern, the sentence in question has now been changed to the following to ensure that the statement is identified as a hypothetical one (line 49-53 in the revised manuscript:

Hypothetically speaking, this could implicate the live attenuated vaccines or nanoparticle-based vaccines as being more effective and convenient due to achieving greater stimulation of the immune response.

6) In addition, why should nanoparticles vaccines be more effective than recombinant or vector based?

We thank the reviewer for this valid query. To address this, the following text has now been included in revised manuscript (line 133-140):

Nanoparticle-based vaccines, such as virus-like particles (VLPs), may also represent a viable and rapid approach, serving as ideal scaffolds for antigen display due to emulation of naturally occurring viruses in size and geometry (Bachmann & Jennings, 2010; Plummer & Manchester, 2010; Frietze et al., 2016;  Shin et al., 2020; Chiba et al., 2021). This results in an enhanced  clustering of B cell receptors and resultant immunogenicity (Bachmann & Zinkernagel, 1997). Indeed, S protein displayed on nanoscale scaffolds seemed more immunogenic in mice compared to S protein administered alone (Zhang et al., 2020), while in hamsters this approach generated high neutralizing antibody titers following a single dose in Syrian hamsters and mice infected with SARS-CoV-2, with no infectious virus detected in the lungs (Chiba et al., 2021; Powell et al., 2021).

7) Line 49 the definition of antibodies as anti-inflammatory agents is not correct, unless the ones that are not against SARS CoV2 are meant. Please specify.

The following text has now been added to address this concern:

Monoclonal antibodies (mAbs) (not against COVID-19) have been utilized as anti-inflammatory agents to combat the cytokine storm characteristic of severe COVID-19.

8) Line 65 pls explain what Ab 4A8 is.

The following has now been added to the manuscript to address the reviewers concern:

The N terminal-domain (NTD) of S-protein also contains immunoprotective epitopes, the most immunodominant of which elicits the production of a mAb with high neutralizing potency against SARS-CoV-2, termed 4A8, without binding receptor binding domain19, 20.

9) Line 86. …interaction leading should be interaction, thus leading.

Changed as suggested

10) Line 93. Sentence missing the verb.

This was a redundant sentence which has now been deleted.

11) Line 100. What are isolates of antibodies?

The sentence has now been changed to the following to avoid confusion:

Neutralizing antibodies targeting the viral nucleocapsid (N) protein, which may mediate viral genome expression and assembly, have been isolated from convalescent sera of COVID-19 patients in the early recovery phase. Of these, anti-S-protein antibodies dominate late phase sera 20, 28, 29.

12) Line 107. Vaccines also stimulate T cells. They are not necessarily made up of a single epitope, I believe the Authors meant antigen not epitope.

Changed as suggested

13) Line 132-135. Sentence requires revision.

The sentence has now been revised to the following:

A placebo-controlled phase 1-2 trial involving 131 healthy adults compared the reactogenicity and safety of anti-S protein IgG titers generated from administration of NVX-CoV2373 to convalescent serum from symptomatic COVID-19 patients.

14) Fig.1 The variants of interest shown in Fig 1 are classidfied of concern by CDC. Pls insert the delta variant. In addition, please add the Greek letters that are used to identify all variants according to CDC.

We thank the reviewer for correctly identifying this and indicating as such. At the time of submission, the old nomenclature was employed, with the Greek letters being assigned after our article was submitted. We have now replaced the old nomenclature with the new Greek letters assigned by the WHO.

15) Lines 173- 176 verb missing.

The sentence has now been changed to the following:

In contrast, the B.1.351 lineage, in addition to displaying E484K and K417N spike mutations, has been shown to confer resistance to neutralization by monoclonal antibody therapy as well as convalescent sera and sera of individuals vaccinated with the Pfizer-BioNTech and Moderna mRNA vaccines.

16) Line 206 inhibition by sera…

Changed as suggested

17) Lines 241-244. Pls rephrase. Again, why do the Authors believe that these formats of vaccination (especially nanoparticles) should work better? Pls explain.

Changed as suggested. Further detail regarding nanoparticle-mediated vaccination has also been addressed in response to a previous query. The following section has been added:

Nanoparticle-based vaccines, such as virus-like particles (VLPs), may also represent a viable and rapid approach, serving as ideal scaffolds for antigen display due to emulation of naturally occurring viruses in size and geometry (Bachmann & Jennings, 2010; Plummer & Manchester, 2010; Frietze et al., 2016;  Shin et al., 2020; Chiba et al., 2021). This results in an enhanced  clustering of B cell receptors and resultant immunogenicity (Bachmann & Zinkernagel, 1997). Indeed, S protein displayed on nanoscale scaffolds seemed more immunogenic in mice compared to S protein administered alone (Zhang et al., 2020), while in hamsters this approach generated high neutralizing antibody titers following a single dose in Syrian hamsters and mice infected with SARS-CoV-2, with no infectious virus detected in the lungs (Chiba et al., 2021; Powell et al., 2021).

Reviewer 2 Report

This article is already somewhat out of date, draws incorrect inferences from cited articles, and offers conclusions that are either totally obvious or unsupported. Amongst the most egregious are:

Line 32  “since antiviral drugs could take years to be widely available” Several promising antivirals are already in advanced testing.

Lines 40-42 “On this basis, we propose that continuous monitoring  for new SARS-CoV-2 sequences based on which mAbs and modifications to vaccines should be developed.” This is already ongoing and a major emphasis of most public health authorities.

Lines 42-44 “Future mRNA and viral-vector COVID vaccines also must account for prevalent SARS-CoV-2 variants. Alternatively, this could implicate the live attenuated vaccines or nanoparticle-based vaccines as being more effective and convenient due to achieving greater stimulation of the immune response.” This is repeated in the conclusion, yet there is no discussion about why this might be so, particularly because neithe of the suggested approaches is as efficacious as the mRNA and even viral vector vaccines. 

Lines 98-101 “ Neutralizing antibodies targeting the viral nucleocapsid (N) protein, which may mediate viral genome expression and assembly, have been isolated from convalescent sera of COVID-19 patients in the early recovery phase; isolates of anti-S-protein antibodies dominate late phase sera 20, 28, 29.” None of the cited references indicated that anti-N Abs are neutralizing.

Throughout the paper and the figure, the old nomenclature for VOCs is used. More recently the VOCs have been assigned greek letters. These should be used.

Lines 170-172 “Analysis of each lineage revealed that the B.1.1.7 lineage discovered in the UK is refractory to several monoclonal antibodies targeting the RBD and NTD of the SARS-CoV- 2 S protein.” In fact the reference cited says “Here we show that B.1.1.7 is refractory to neutralization by most monoclonal antibodies against the N-terminal domain of the spike protein and is relatively resistant to a few monoclonal antibodies against the receptor-binding domain.”

Lines 238-240: “Lastly, considering future potentially resistant strains as 238 well as the ability to vaccinate the majority of the growing human population, live attenuated vaccines could constitute better options than mRNA and viral vector vaccines from a convenience standpoint” There is no evidence to support this statement, particularly in view of the fact that such vaccines have lower efficacy than the RNA vaccines. While there might be theoretical advantages to using whole virus as an immunogen (eg increased targets for CTLs), no rationale is given for their proposal.

Author Response

Reviewer 2:

1) Line 32  “since antiviral drugs could take years to be widely available” Several promising antivirals are already in advanced testing.

The reviewer is indeed correct in their assertion that antivirals are currently in advanced testing, and it was not our purpose to indicate that antiviral drugs are not a viable option. We meant to indicate that in terms of a rapid, short-term solution, antiviral drugs may not represent the most rapid option. To this degree, we have modified our relevant discussion to the following, and we thank the reviewer for pointing out this misrepresentation:

Antibody therapy, mass vaccination programs, and antiviral drugs are regarded as most effective. However, even though a number of antiviral drugs are currently in advanced testing, the development of new drugs is complex and can be a prolonged process, with repurposed drugs currently representing an alternative (Venkadapathi et al., 2021). However, while repurposing current drugs is promising, viral plasticity renders repurposed drugs against different targets than SARS-CoV-2 with a relatively limited shelf-life (Dolgin, 2021).  Thus, while recent efforts are indeed encouraging, it may take some time for suitable and widely effective antiviral drugs to become globally available. Until then, vaccination appears to be the most currently rapidly feasible option.

2) Lines 40-42 “On this basis, we propose that continuous monitoring  for new SARS-CoV-2 sequences based on which mAbs and modifications to vaccines should be developed.” This is already ongoing and a major emphasis of most public health authorities.

The sentence in question has now been modified to the following statement. We thank the reviewer for pointing out and correcting this misrepresentation:

Indeed, continuous monitoring for new SARS-CoV-2 sequences based on which mAbs and modifications to vaccines should be developed is currently an ongoing emphasis of most public health authorities.

3) Lines 42-44 “Future mRNA and viral-vector COVID vaccines also must account for prevalent SARS-CoV-2 variants. Alternatively, this could implicate the live attenuated vaccines or nanoparticle-based vaccines as being more effective and convenient due to achieving greater stimulation of the immune response.” This is repeated in the conclusion, yet there is no discussion about why this might be so, particularly because neithe of the suggested approaches is as efficacious as the mRNA and even viral vector vaccines. 

The reviewer indeed makes a valid point, as this was not fully discussed in the previous version of the manuscript, and was a point raised by the other reviewer as well. To this degree, we have added the following items of discussion, at various parts of the manuscript, to address both concerns:

Nanoparticle-based vaccines, such as virus-like particles (VLPs), may also represent a viable and rapid approach, serving as ideal scaffolds for antigen display due to emulation of naturally occurring viruses in size and geometry (Bachmann & Jennings, 2010; Plummer & Manchester, 2010; Frietze et al., 2016;  Shin et al., 2020; Chiba et al., 2021). This results in an enhanced  clustering of B cell receptors and resultant immunogenicity (Bachmann & Zinkernagel, 1997). Indeed, S protein displayed on nanoscale scaffolds seemed more immunogenic in mice compared to S protein administered alone (Zhang et al., 2020), while in hamsters this approach generated high neutralizing antibody titers following a single dose in Syrian hamsters and mice infected with SARS-CoV-2, with no infectious virus detected in the lungs (Chiba et al., 2021; Powell et al., 2021).

Rather encouragingly, concerns underlying variants that could be partially resistant to vaccine-induced antibody responses may exhibit other immune responses that protect against viruses. Of particular interest is T-cell-mediated immunity, which targets and eliminates virus-infected cells. Indeed, data is increasingly suggesting that such responses could provide enhanced immunity against COVID-19 despite decreased effectiveness of antibody-based responses (Ledford, 2021; Skelly et al., 2021; Tarke et al., 2021). While T cells do not prevent infection, such cells exert a role in clearing an infection, potentially meaning the  difference between a mild or severe infection, potentially also reduce community transmission (Ledford, 2021; Tarke et al., 2021). Furthermore, T cells could also be more resistant to decreased efficacy towards emerging variants compared to antibodies (Ledfor, 2021). Indeed, T cell responses to coronavirus vaccination or previous infection did not target mutated regions found in variants, conferring a type of ‘resistance’against such mutations (Ledford, 2021; Tarke et al., 2021). While most antibody-based vaccines target the mutation-prone S region of the virus, T cells target very stable viral proteins inside infected cells, potentially indicating powerful avenues of vaccination research incorporating targets from multiple proteins into one vaccine (Ledford, 2021).

Indeed, clinical trials evaluating efficacy of the inactivated Sinopharm and Sinovac vaccines have revealed a lower efficacy than the mRNA vaccines against the Wuhan reference strain as well as the VOCs (1,2). However, due to favorable storage conditions (2–8 degrees Celsius), these vaccines are in widespread use in middle- to low-income countries  (3). However, nanoparticle vaccines (also termed nanovaccines) such as the current Novavax vaccine could be more feasible from an efficacy and safety standpoint. Indeed, numerous antigens can be incorporated into nanoparticles either via entrapment (nanoparticle engulfs and thereby protects the antigen) or conjugation (covalent linkage), yielding robust immunogenicity (4). Furthermore, certain nanoparticles can independently act as adjuvants by inducing reactive oxygen species production in APCs, leading to NLPR3 inflammasome activation (5) and augmentation of the immune response through neutrophil recruitment and NETosis (6–8). Indeed, a polymerase-based nanovaccine delivering the SARS-CoV-2 S-protein RBD elicited a robust immunological response (9). In short, nanoparticle-based vaccines show no compromise in immunogenicity, favoring their use in vaccines.

The main advantage of nanoparticles is that due to their size and ability to incorporate viral antigens, these vaccines can mimic respiratory viruses and their pathophysiology. Further, recent advancements in this field have enabled tailoring of nanoparticles with specific shapes, solubility, surface chemistry, and hydrophilicity (10,11). SARS-CoV-2 and other respiratory viruses are transmitted via droplets and/or aerosols, subsequently multiplying and invading the respiratory mucosa all while simultaneously propagating along the conducting airways (ciliated respiratory epithelial cells) (12,13). The current COVID vaccines are administered intramuscularly, which while eliciting a systemic immune response, does not induce protective mucosal antigen-specific IgA antibodies. On the other hand, nanoparticle-based vaccines do not suffer from such limitations, and can mimic virus behavior. Indeed, intranasal routes of administration confer better protection, i.e., on both systemic and mucosal levels (14–18) . Consistently, studies evaluating efficacy of nanoparticles against respiratory viruses include influenza, respiratory syncytial virus, and Bovine parainfluenza 3 viruses have yielded encouraging results [refer to (11) for detailed review].

Admittedly, however, several challenges do exist in the production of such vaccines, mainly related to production difficulties (19). Furthermore, the interactions between nanoparticles as adjuvants and immune cells remain poorly understood (11,20,21). Nevertheless, nanoparticle-based vaccines merit further investigation and scrutiny as validated strategies. Despite the production difficulties, the effectiveness of nanoparticle-based vaccines in combating the aforementioned respiratory viruses could translate over to SARS-CoV-2; intranasally administered nanoparticle-based vaccines could represent a feasible preventative modality.

4) Lines 98-101 “ Neutralizing antibodies targeting the viral nucleocapsid (N) protein, which may mediate viral genome expression and assembly, have been isolated from convalescent sera of COVID-19 patients in the early recovery phase; isolates of anti-S-protein antibodies dominate late phase sera 20, 28, 29.” None of the cited references indicated that anti-N Abs are neutralizing.

The reviewer is correct in their assertion. This was a typographical error, and we wanted to indicte presence of such antibodies and not whether they were neutralizing or not. The sentence has now been modified accordingly.

5) Throughout the paper and the figure, the old nomenclature for VOCs is used. More recently the VOCs have been assigned greek letters. These should be used.

We thank the reviewer for correctly identifying this and indicating as such. At the time of submission, the old nomenclature was employed, with the Greek letters being assigned after our article was submitted. We have now replaced the old nomenclature with the new Greek letters assigned by the WHO.

6) Lines 170-172 “Analysis of each lineage revealed that the B.1.1.7 lineage discovered in the UK is refractory to several monoclonal antibodies targeting the RBD and NTD of the SARS-CoV- 2 S protein.” In fact the reference cited says “Here we show that B.1.1.7 is refractory to neutralization by most monoclonal antibodies against the N-terminal domain of the spike protein and is relatively resistant to a few monoclonal antibodies against the receptor-binding domain.”

While the reviewer correctly quotes the reference in this instance, perhaps our mode of communication was incomplete as we consider we had said more or less the same thing as the reference quoted. We had meant to purport that the B.1.1.7 lineage seemed more resistant to neutralization by antibodies against both the RBC and NTD. Perhaps the reviewer would like us to be more specific in the relative amount of neutralization by the antibodies? Regardless, we have now used the same terms as used in the reference itself:

Analysis of each lineage revealed that the B.1.1.7 lineage discovered in the UK was resistant to neutralization by most mAbs against the NTD of the S protein, and was also relatively resistant to a few mAbs against the RBD.

7) Lines 238-240: “Lastly, considering future potentially resistant strains as 238 well as the ability to vaccinate the majority of the growing human population, live attenuated vaccines could constitute better options than mRNA and viral vector vaccines from a convenience standpoint” There is no evidence to support this statement, particularly in view of the fact that such vaccines have lower efficacy than the RNA vaccines. While there might be theoretical advantages to using whole virus as an immunogen (eg increased targets for CTLs), no rationale is given for their proposal.

The reviewer is correct in this assertion, and we thank them for this very pertinent suggestion. To address this concern, we have now added the following segment to our manuscript:

Indeed, clinical trials evaluating efficacy of the inactivated Sinopharm and Sinovac vaccines have revealed a lower efficacy than the mRNA vaccines against the Wuhan reference strain as well as the VOCs (1,2). However, due to favorable storage conditions (2–8 degrees Celsius), these vaccines are in widespread use in middle- to low-income countries  (3). However, nanoparticle vaccines (also termed nanovaccines) such as the current Novavax vaccine could be more feasible from an efficacy and safety standpoint. Indeed, numerous antigens can be incorporated into nanoparticles either via entrapment (nanoparticle engulfs and thereby protects the antigen) or conjugation (covalent linkage), yielding robust immunogenicity (4). Furthermore, certain nanoparticles can independently act as adjuvants by inducing reactive oxygen species production in APCs, leading to NLPR3 inflammasome activation (5) and augmentation of the immune response through neutrophil recruitment and NETosis (6–8). Indeed, a polymerase-based nanovaccine delivering the SARS-CoV-2 S-protein RBD elicited a robust immunological response (9). In short, nanoparticle-based vaccines show no compromise in immunogenicity, favoring their use in vaccines.

The main advantage of nanoparticles is that due to their size and ability to incorporate viral antigens, these vaccines can mimic respiratory viruses and their pathophysiology. Further, recent advancements in this field have enabled tailoring of nanoparticles with specific shapes, solubility, surface chemistry, and hydrophilicity (10,11). SARS-CoV-2 and other respiratory viruses are transmitted via droplets and/or aerosols, subsequently multiplying and invading the respiratory mucosa all while simultaneously propagating along the conducting airways (ciliated respiratory epithelial cells) (12,13). The current COVID vaccines are administered intramuscularly, which while eliciting a systemic immune response, does not induce protective mucosal antigen-specific IgA antibodies. On the other hand, nanoparticle-based vaccines do not suffer from such limitations and can mimic virus behavior. Indeed, intranasal routes of administration confer better protection, i.e., on both systemic and mucosal levels (14–18) . Consistently, studies evaluating efficacy of nanoparticles against respiratory viruses include influenza, respiratory syncytial virus, and Bovine parainfluenza 3 viruses have yielded encouraging results [refer to (11) for detailed review].

Admittedly, however, several challenges do exist in the production of such vaccines, mainly related to production difficulties (19). Furthermore, the interactions between nanoparticles as adjuvants and immune cells remain poorly understood (11,20,21). Nevertheless, nanoparticle-based vaccines merit further investigation and scrutiny as validated strategies. Despite the production difficulties, the effectiveness of nanoparticle-based vaccines in combating the respiratory viruses could translate over to SARS-CoV-2; intranasally administered nanoparticle-based vaccines could represent a feasible preventative modality.

Round 2

Reviewer 2 Report

My major criticisms have been addressed. However, there is still the question of whether this review actually adds anything to this rapidly changing field.